# Anti-Biofilm Activity of a Hyaluronan-like Exopolysaccharide from the Marine *Vibrio* MO245 against Pathogenic Bacteria

**DOI:** 10.3390/md20110728

**Published:** 2022-11-21

**Authors:** Marie Champion, Emilie Portier, Karine Vallée-Réhel, Isabelle Linossier, Eric Balnois, Guillaume Vignaud, Xavier Moppert, Claire Hellio, Fabienne Faÿ

**Affiliations:** 1Laboratoire de Biotechnologies et Chimie Marines, EA3884, Université Bretagne Sud, 56321 Lorient, France; 2Laboratoire de Biotechnologies et Chimie Marines, EA3884, Université de Bretagne Occidentale, 29000 Quimper, France; 3Institut de Recherche Dupuy de Lôme (IRDL), UMR CNRS 6027, Université Bretagne Sud, 56321 Lorient, France; 4Pacific Biotech, BP 140 289, 98701 Arue, France; 5Biodimar, LEMAR UMR6539, Institut Européen de la Mer, Université Bretagne Occidentale, 29238 Brest, France

**Keywords:** exopolysaccharide, anti-adhesive activity, anti-biofilm activity, marine bacteria, non biocidal agent, *Pseudomonas aeruginosa*, *Vibrio harveyi*

## Abstract

Biofilms, responsible for many serious drawbacks in the medical and marine environment, can grow on abiotic and biotic surfaces. Commercial anti-biofilm solutions, based on the use of biocides, are available but their use increases the risk of antibiotic resistance and environmental pollution in marine industries. There is an urgent need to work on the development of ecofriendly solutions, formulated without biocidal agents, that rely on the anti-adhesive physico-chemical properties of their materials. In this context, exopolysaccharides (EPSs) are natural biopolymers with complex properties than may be used as anti-adhesive agents. This study is focused on the effect of the EPS MO245, a hyaluronic acid-like polysaccharide, on the growth, adhesion, biofilm maturation, and dispersion of two pathogenic model strains, *Pseudomonas aeruginosa* sp. PaO1 and *Vibrio harveyi* DSM19623. Our results demonstrated that MO245 may limit biofilm formation, with a biofilm inhibition between 20 and 50%, without any biocidal activity. Since EPSs have no significant impact on the bacterial motility and quorum sensing factors, our results indicate that physico-chemical interactions between the bacteria and the surfaces are modified due to the presence of an adsorbed EPS layer acting as a non-adsorbing layer.

## 1. Introduction

In the natural environment, bacteria do not live in a planktonic condition but as a biofilm [1]. This way of life provides microorganisms with better resistance to environmental stress. Several steps govern the formation of biofilms: reversible and then irreversible adhesion, cell multiplication, the formation of micro colonies, and a polymeric extracellular matrix, the overall process forming what is called biofilm [2,3,4]. This matrix allows the bacteria to exchange genetic information, access a supply of nutrients, and resist antibiotics as well as allowing for intercellular communication (quorum sensing) and defense mechanisms [1,5,6].

Biofilm formation and increased resistance to environmental stresses cause many problems in marine and medical environments. In the medical environment, any surface, biological or not, is vulnerable to colonization by bacteria. The increase in infection problems in hospitals is partly due to new technologies that use more and more medical technologies and medical equipment. Indeed, the formation of biofilm on this equipment leads to more interventions (removing the contaminant), treatments (antibiotics), and antibiotic resistance. Moreover, these issues also lead to higher economic costs [7,8]. Infections due to biofilms are responsible for at least 80% of all infections, representing a significant problem for human health [9]. According to the World Health Organization, approximately 700,000 people worldwide die each year because of resistance to the various existing anti-microbial treatments [10].

In the marine environment, the main problem related to biofilm is the development of biofouling on immersed surfaces. Biofouling is characterized by the first stage of microbial colonization (bacteria and diatoms) called microfouling, prior to the adhesion and colonization of macroorganisms such as algae, barnacles, and mussels. Biofouling leads to a loss of structural integrity and performance, corrosion [11], and the settlement and development of unwanted aquatic species. Biofouling communities may compete with cultivated organisms, including predators and host diseases. Several diseases in marine vertebrates and invertebrates are caused by the ability of pathogens to attach and form biofilms, leading to vasculitis, gastroenteritis, and ocular lesions in fish. The economic impact of biofouling management in aquaculture is estimated to be 5–10% of production costs [12]. Furthermore, the use of antibiotics in aquaculture can lead to the spread of antibiotic residues in the marine environment, increasing the rate of antibiotic resistance in marine bacteria and ultimately transferring this resistance to human pathogens [13].

Control methods, such as antibiotics and biocides, are confronted with problems of antibiotic resistance and pollution. In fact, the regulation of biocidal products is in constant evolution with improving commercial solution quality to ensure a high level of protection of human health and the environment as well as antifouling efficacy [14].

This is why it is necessary to be proactive and actively work on the discovery of new alternatives to these agents. One line of research is the use of active molecules to inhibit adhesion without toxicity. This is particularly the case for EPSs, which are known as anti-biofilm agents. For example, marine bacterial EPSs have shown anti-biofilm activity against *Pseudomonas aeruginosa*, *Vibrio* sp., *Escherichia coli*, and *Staphylococcus aureus* [15,16,17,18]. EPSs have the ability to inhibit and/or disrupt biofilm formation through different modes of action [19]. They are able to modify the physico-chemistry of biotic or abiotic surfaces, they can modulate the expression of cell-cell communication genes or adhesion factors, and they also have the ability to act on biofilms thanks to biosurfactant properties [19]. For example, EPS273, produced by a marine bacterium, *Pseudomonas stuzeri*, plays an anti-biofilm role in regulating the expression of the PhoP-PhoQ gene and in the quorum sensing pathway [20]; in addition, the EPS A101 from a *Vibrio* strain was able to disperse *P. aeruginosa* biofilm due to the potential EPS/bacterial surface interaction [16].

MO245 is an EPS produced by a marine bacterium, a *Vibrio alginolyticus* sp., isolated from microbial mats located in Moorea Island lagoon (French Polynesia). This EPS has a molecular weight of 1.5 MDa and is composed of repeating units of glucuronic acid, N-acetylglucosamine, and N-acetyl-galactosamine, 2:1:1 (→4)-β-D-GlcpA-(1→4)-α-D-GalpNAc-(1→3)-β-D-GlcpNAc-(1→4)-β-D-GlcpA-(1→). Due to its structure and rheological properties (Figure 1), MO245 shows similarities with hyaluronic acid (HA), which may lead to some close physico-chemical properties [21]. HA (1.5 MDa) is composed of D-glucuronic acid and D-N-acetyl-glucosamine, 1:1 (→4)-β-D-GlcpA- (1→3)-β-D-GlcpNAc-(1→). Both of them are linear polymers with alternating 1→3 and 1→4 linkages [22]. Moreover, HA has shown interesting anti-adhesive properties [23], without bactericidal effects [24].

The activity of MO245 was studied on two bacteria. The first one was *Pseudomonas aeruginosa* (PaO1), a bacterium that is often associated with biofilm-related infections. This strain plays an important role in infections such as cystic fibrosis pneumonia, chronic wound infection, chronic otitis media, chronic bacterial prostatitis, and medical-related infection [25]. The second one was *Vibrio harveyi* (DSM19623), a marine bacterial model. It is a pathogenic bacterium of wild and aquaculture fish and marine invertebrates. Diseases caused by this bacterium have lead to serious economic problems in China, Japan, Europe, and North America [26].

The aim of this work is (i) to evaluate the anti-adhesive and anti-biofilm activity of the hyaluronan-like MO245 compared to HA against pathogenic bacteria, and (ii) to determine its mode of action.

## 2. Results

### 2.1. Rheological Characteristics

According to the literature, MO245 and HA were described as having non-Newtonian or pseudoplastic behavior at high concentrations, with shear-thinning and elastoviscous behavior. A clear relationship between viscosity and concentration of polysaccharide solutions has been previously highlighted for MO245 and HA [21,27].

At equal concentrations, MO245 has a lower viscosity than HA (Table 1). The elastic (G′) and viscous (G″) moduli of MO245 are always lower than HA in the same concentrations and at the same frequency.

### 2.2. Biological Activity of MO245 and HA

#### 2.2.1. Anti-Bacterial Activity

1.Bacteriostatic activity

The bacteriostatic activity of MO245 and HA on *P. aeruginosa* and *V. harveyi* was evaluated in a rich medium. Several concentrations were tested: 125, 250, and 500 µg/mL (full data set not shown). The impact on the growth of bacteria was based on a comparison between the kinetics in the control Luria Bertani (LB) or Zobell medium for *P. aeruginosa* and *V. harveyi*, respectively, without or with the addition of MO245 or HA in this medium.

Figure 2A,B shows that MO245 and HA did not have any impact on the growth curve of *P. aeruginosa* and *V. harveyi*. The result for all concentrations was the same.

2.Use of MO245 or HA as a carbon source

MO245 and HA were tested as a carbon source in an M9 minimum medium supplemented with MO245 or HA at 125 µg/mL. The results were compared with glucose as a carbon source at 4 g/L (Figure 2C,D).

The cell concentration did not significantly increase over time in the presence of MO245 or HA. In contrast, in the presence of glucose, the concentration increased from 10^8^ to 10^9^ CFU/mL over the same period. The bacteria did not use MO245 or HA at 125 µg/mL as a carbon source.

3.Bactericidal effect

The bactericidal activity of MO245 and HA was tested in minimum medium M9 supplemented with MO245 or HA at 125 µg/mL. The results were compared to a condition without MO245 or HA (Figure 2C,D).

The cell concentration remained at 10^8^ CFU/mL and did not significantly decrease over time in presence of MO245 or HA. MO245 and HA at 125 µg/mL have no bactericidal effect as shown in Figure 2C,D.

The concentration of 125 µg/mL, the minimum concentration with no bacteriostatic effect, was chosen for the rest of the studies in order to study their impact on adhesion and the biofilm.

#### 2.2.2. Anti-Adhesion Activity

MO245 and HA activity on the adhesion of *P. aeruginosa* and *V. harveyi* was evaluated in a flow cell system [28]. The activity of MO245 and HA was assessed in two conditions (Figure 3).

The first condition was the addition of MO245 or HA within the minimum bacterial media, allowing the study of the interactions with the biotic surface (called the first condition). The second was conditioning the adhesion glass surface slide to study the interactions with the abiotic surface (referred to as the second condition).

Table 2 shows the impact of MO245 and HA on the adhesion of *P. aeruginosa* and *V. harveyi*. For *P. aeruginosa*, the addition of MO245 at 125 µg/mL within the minimum bacterial medium shows a significant (*p* < 0.001) 53% decrease in adhesion. The effect of MO245 in the second condition showed a significant decrease (*p* < 0.001) of 44% of adhesion. For *V. harveyi*, MO245 decreased adherence by 29 and 49% (*p* < 0.001) compared to the control when there was a conditioning of the biotic (first condition) and abiotic surface (second condition).

In all cases, the activity of MO245 was significantly (*p* < 0.001) more than HA at 125 µg/mL. HA did not demonstrate an impact on the adhesion of either bacterial strain.

#### 2.2.3. Impact on the Biofilm Maturation

The impact of MO245 and HA at 125 µg/mL on biofilm maturation was tested on *P. aeruginosa* and *V. harveyi*. Biofilms were grown after allowing 2 h for bacterial adhesion in the absence of MO245 or HA, and then a 24 h supplemented flow to allow colonization.

The addition of MO245 at 125 µg/mL in the growth medium led to a significant decrease in the *P. aeruginosa* biofilm maturation. Figure 4 shows a reduction in the biomass of 38% (*p* < 0.001). The average thickness decreased by 26% (*p* < 0.001) compared to the control. However, with the same condition, HA had no significant impact (*p* < 0.05) on biofilm maturation.

For the *V. harveyi* strain, Figure 5 shows that the addition of MO245 or HA at 125 µg/mL significantly decreased biofilm maturation. Unlike the case with *P. aeruginosa*, the addition of HA in the growth medium led to a significant (*p* < 0.01) decrease of 38% of the biomass and 37% (*p* < 0.01) in the average thickness of the biofilm. MO245 led to a significant (*p* < 0.001) diminution of 53% of the biomass and a significant (*p* < 0.001) diminution of 49% of the average thickness.

A double staining with Syto9^®^ and propidium iodide was performed. The results showed no red cells, therefore no dead cells (data not shown).

#### 2.2.4. Impact on the Degradation of the Biofilm

The impact of MO245 and HA was tested on the disruption of a biofilm already formed. *P. aeruginosa* or *V. harveyi* biofilms were formed in the flow cell system for 24 h in a growth medium not supplemented with MO245 or HA. After 24 h of incubation, MO245 or HA at 125 µg/mL were inoculated in the flow cell chamber. The degradation was observed after 2 h of MO245 or HA incubation on the biofilm and 30 min of washing.

The impact of MO245 and HA on the disruption of *P. aeruginosa* biofilm (Figure 6) showed that MO245 and HA significantly decreased the average thickness by approximately 23%. However, only MO245 significantly decreased the biomass of the biofilm by 19% (*p* < 0.01).

Figure 7 shows that the effect of MO245 and HA on the preformed *V. harveyi* biofilm led to the same results as *P. aeruginosa*. MO245 and HA significantly decreased (*p* < 0.01) the average thickness by 30% and 27%, respectively. As previously shown for *P. aeruginosa*, only MO245 had an impact on the biomass: the decrease was 28% (*p* < 0.01).

A double staining with Syto9^®^ and propidium iodide was performed. The results showed no red cells, therefore no dead cells (data not shown).

#### 2.2.5. Morphological Impact on Bacteria

To understand how MO245 and HA could act on the cells, the impact on morphology was tested. The cells were adhered to glass coverslips with and without conditioning for 2 h at 20 °C in minimum medium and then observed, after fixation with glutaraldehyde and ethanol dehydration, under a scanning electron microscope (SEM). No impact on morphology was observed in the presence of MO245 or HA with and without conditioning (Appendix A). MO245 and HA activity would not appear to be from a change in cell morphology.

### 2.3. Evaluation of the Biological Role of MO245 and HA

#### 2.3.1. Impact on the Bacterial Motility

MO245 or HA was incubated at 125 µg/mL with *P. aeruginosa* or *V. harveyi*, in physiological water or artificial seawater (ASW), respectively, for 2 h. The impact of MO245 and HA on the motility was shown with the measurement of a growing circular zone. Separate agar plates were used for swimming, swarming, and twitching tests [29].

Figure 8A shows the impact of HA on the swarming of *P. aeruginosa* and the impact of MO245 on the twitching of the bacteria. However, no impact on the swimming was recorded. For *V. harveyi*, no impact was recorded from MO245 or HA on the motility of the bacterium as shown in Figure 8B.

The impact of MO245 or HA on *P. aeruginosa* and *V. harveyi* does not appear to be linked to a decrease in the motility of bacteria.

#### 2.3.2. Anti-Quorum Sensing Properties

The *E. coli* biosensor pSB401 allows the recognition of C6-HSLs leading to luminescence emission by activation of the *lux*CDABE gene. This luminescence is directly correlated to the quorum sensing of *P. aeruginosa*. The impact of MO245 and HA on the quorum sensing of *P. aeruginosa* has been studied by this mean. Kojic acid, a known inhibitor of quorum sensing, was used as a control.

The results (Figure 9A) showed no significant effect of MO245 or HA on the luminescence to optical density ratio (RLU) of the *E. coli* pSB401 biosensor. Therefore, MO245 and HA had no impact on this quorum-sensing pathway of *P. aeruginosa* using C6-HSLs.

*V. harveyi* is a bioluminescent bacterium. The bioluminescence is directly related to the quorum sensing of the bacterium [30]. The impact of MO245 and HA was therefore tested on the quorum sensing of *V. harveyi* by a kinetic of the RLU. The results (Figure 9B) showed no effect of MO245 and HA at 125 µg/mL on *V. harveyi* quorum sensing.

Additionally, no decrease in luminescence was observed over time in the presence of the MO245 or HA, in contrast to kojic acid.

### 2.4. Evaluation of Cell-Surface Interactions

#### 2.4.1. Microbial Adherence to Hydrocarbons (MATH)

The MATH test allows the study of the impact of MO245 and HA on the surface hydrophobicity of bacteria [31]. For this purpose, *P. aeruginosa* and *V. harveyi* were exposed to MO245 or HA for 2 h and then toluene. Estimation of the hydrophobicity was performed by measuring the absorbance of the aqueous phase.

*P. aeruginosa* showed a hydrophobicity of 62% and *V. harveyi* of 76%. These results were supported by previous studies on *P. aeruginosa* and *V. harveyi* showing a surface hydrophobicity of 68% [32] and 80%, respectively [30].

Results, Figure 10, showed that MO245 significantly decreased the surface hydrophobicity of *P. aeruginosa* from 62% to 34% (*p* < 0.05), in contrast to HA. However, the MATH assay did not show a change in the hydrophobicity of *V. harveyi* in the presence of MO245 or HA. The hydrophobicity of *V. harveyi* remained at approximately 76%.

The impact of MO245 on the adhesion and biofilm of *P. aeruginosa* could come from the modification of the hydrophobicity of the cells. This method did not lead to the same conclusion for *V. harveyi*.

#### 2.4.2. Quartz Crystal Microbalance (QCM) Measurements

The adsorption kinetic of MO245 and HA was investigated by QCM, under the same pH and salinity conditions as for bacterial adhesion. Figure 11 represents the shift in frequency vs. time for MO245 and HA. For HA, no shift of frequency could be observed, revealing that HA did not adsorb on the SiO_2_ surface under these conditions. This result was in accordance with previous studies [33]. On the contrary, for MO245, a decrease in frequency up to −18 Hz was recorded, indicating the formation of an adsorbed layer on the surface. The adsorption of other EPS, under similar conditions (pH 6.2 and I = 100 mM NaCl) has already been observed by QCM [34], proving the complex adsorption behavior of MO245 toward the SiO_2_ surface in comparison with HA.

#### 2.4.3. Water Contact Angle of the Abiotic Surface Interaction with MO245 or HA

The water contact angle was used to measure the hydrophobicity of a surface. To determine the impact of MO245 and HA on the change in hydrophobicity of a surface, the contact angle with water was measured on surfaces conditioned with MO245 or HA.

A decrease in the contact angle by about 40° (*p* < 0.001) was recorded when the glass surface was conditioned with MO245 and HA, leading to a hydrophilic surface (Figure 12).

#### 2.4.4. Emulsifying Properties

In order to efficiently study the surface active properties of MO245 and HA quickly and simply, the emulsion index was calculated and the stabilization property of the emulsion over time was measured [35].

For this purpose, MO245 or HA was mixed with olive oil as an apolar phase. The emulsion layer was measured at 1 h, 24 h, 48 h, and 168 h, to determine the emulsion index over time. A known surfactant, Triton X-100, was chosen as a positive control.

The results, Figure 13, showed that MO245 and HA are able to create an emulsion with olive oil. The results also highlight that they had a lower emulsion index than Triton X-100, around 60% for MO245 and HA compared to 70% for Triton X-100. However, they were able to keep the emulsion stable at around 60% up to 168 h. Both MO245 and HA had interesting surface-active properties.

## 3. Discussion

EPSs have displayed interesting anti-adhesive and anti-biofilm properties in many studies [19] related to their biological activity or their physico-chemical impact [19,36]. This study evaluated (i) the anti-adhesive and anti-biofilm properties of an EPS, MO245, to understand its mechanisms of action from a (ii) biological as well as a (iii) physicochemical point of view.

### 3.1. Proven Anti-Biofilm Activity for MO245

MO245 and HA, at 125 µg/mL, did not show bactericidal or bacteriostatic effects against *P. aeruginosa* and *V. harveyi* strains. As with MO245, this result has already been found for several EPS in the literature. For example, a marine bacterium, *Bacillus licheniformis* T14, produced a 1000 kDa EPS with no effect on growth at 400 µg/mL [18]. However, HA has already shown a strain and concentration-dependent bacteriostatic effect [24].

HA tested at 125 µg/mL, did not show anti-biofilm activity against *P. aeruginosa*. Nevertheless, on *V. harveyi*, HA resulted in a decrease in biofilm biomass. HA has already been described as a polysaccharide with variable anti-biofilm properties depending on the molecular weight, concentration, and strains [37]. MO245 was able to disrupt the adhesion of *P. aeruginosa* and *V. harveyi*, prevent biofilm formation, and degrade biofilm, thus showing anti-biofilm activity. This EPS displayed an interesting activity without toxicity as already described in other EPSs. An EPS produced by a marine bacterium, *Pseudoalteromonas* NCIMB 2021, showed anti-adhesive activity by adsorbing onto a chromium surface [38]. *B. licheniformis* EPS exhibited dose-dependent anti-biofilm effects on four different strains, including *P. aeruginosa* [30].

Several EPSs have already shown similarities in monosaccharide composition and molecular weight without having the same viscoelasticity, however, no obvious correlation has been determined between these three parameters. One of the hypotheses concerning the difference in the anti-adhesive properties between MO245 and HA could come from the differences in their viscoelastic properties [39].

### 3.2. The Biological Role of Quorum Sensing of MO245 Was Not Demonstrated

Quorum sensing is a form of cellular communication involving signaling molecules, autoinducers, and receptors [40]. It is an important mechanism in pathogenic Gram-negative bacteria for the production of virulence factors and the formation of biofilms [41].

In *P. aeruginosa* and *V. harveyi*, quorum sensing has been well-studied and widely described [42,43,44]. Many studies aimed to disrupt this communication pathway to fight against bacterial biofilm formation and proliferation [45].

Therefore, in order to understand how MO245 reduces *P. aeruginosa* and *V. harveyi* biofilm, anti-quorum sensing assays were performed. Results showed no impact of MO245 on one of the quorum sensing paths of *P. aeruginosa* and *V. harveyi*. Nevertheless, many active EPS showed biological activity by targeting quorum sensing and virulence factors, impacting the expression of certain genes and competitively inhibiting the interactions involved in adhesion [19,46]. For example, *S. thermophilus* produces an EPS with the ability to inhibit the production of the violacein pigment of the *C. violaceum* biosensor and the production of the C6-HSL signal molecule. These results highlighted the effect of EPSs on the expression of virulence and quorum-sensing genes of pathogenic microorganisms [47]. Other EPSs, such as EPS 273 produced by *P. stutzeri* 273, had effects on the virulence factors of *P. aeruginosa* (pyocyanin), thus reducing H_2_O_2_ production and extracellular DNA excretion [48,49].

However, the observed anti-adhesive and anti-biofilm effects of MO245 did not originate from properties inhibiting the known biological factors of adhesion and biofilm maturation.

### 3.3. Is the Anti-Biofilm Activity of MO245 Due to Its Physico-Chemical Properties?

The physical chemistry of bacterial and abiotic surfaces plays an important role in cell adhesion and thus in the development of biofilms. Various interactions are involved in the early stages of adhesion and hydrophobic and electrostatic interactions are the most studied [4]. Indeed, bacteria, being negatively charged, preferentially attach themselves to positively charged surfaces. Therefore, negatively charged surfaces should cause electrostatic repulsion, thus decreasing the adhesion of bacteria. Moreover, bacteria adhere better to hydrophobic surfaces, as seen by the favorable interactions of flagella, fimbriae, and pili on these surfaces [3,4,50].

The results of the impact of MO245 on surfaces showed that the EPS is able to (i) significantly decrease the hydrophobicity of *P. aeruginosa* but not *V. harveyi*, and (ii) significantly decrease the hydrophobicity of a glass slide. Many EPSs have already shown their anti-biofilm efficiency thanks to their physico-chemical properties [51]. For example, an EPS produced by *B. licheniformis* had the ability to reduce the surface hydrophobicity of *E. coli* by 60% and of *P. fluorescens* by 25%. Moreover, this EPS also showed anti-biofilm activities on many pathogenic strains without having any effect on growth. The mechanism of action of this activity seems to be independent of an anti-quorum sensing activity but is exerted by the modification of the surfaces by decreasing the hydrophobicity of the cells [30]. In a cyanobacterium (*Synechocystis*), the production of EPS showed a significant decrease in surface hydrophobicity and interesting anti-biofilm properties. Indeed, hydrophobicity tests on the EPS-producing strain showed a significant decrease in surface hydrophobicity of 90% [52]. Similarly to MO245, the EPS Ec300 produced by an *E. coli* strain decreased the hydrophobicity of a glass surface by 20° and reduced the formation of an *S. aureus* biofilm by 40% [53]. Indeed, it is often observed that surfaces with a hydrophilic character were less favorable to adhesion by microorganisms [54].

MO245 via QCM-D confirmed its ability to adsorb to a silica surface, to remain after washing, and thus to change the surface properties as described previously. The ability of EPSs to adsorb to surfaces has already been demonstrated. For example, dextran, an important bacterial extracellular polysaccharide, has been shown to adsorb to a silica surface [55]. However, in the case of MO245, a washing of the quartz crystal resulted in only a slight decrease in the adsorbed layer, reflecting stronger interactions with the surface than in the case of dextran, where it was completely removed.

In addition, MO245 played a role at the water-oil interface due to its emulsifying properties; MO245 was able to stabilize an oil-water emulsion over time. These surfactant properties have already been demonstrated with other EPSs and linked with anti-biofilm activities. At the same concentration as MO245, and after 168 h, two EPSs produced by *B. amyloliquefaciens* LPL061 had an emulsification index of 30% [56]. The *B. licheniformis* T14 EPS displayed a similar emulsion index to MO245 (60%) but at half the concentration [18]. Other EPSs have shown their ability to act as surfactants such as an EPS produced by *L. helveticus* MB2-1 which has the ability to stabilize an emulsion over time and inhibit the attachment step and self-aggregation by decreasing the interactions between cells or cells and a substrate [57]. For an *O. iheyensis* BK6 strain, the anti-biofilm properties come from both surface tension reduction and emulsifying properties [58].

### 3.4. Complex Physico-Chemical Properties of MO245

In this study, the activity of MO245 was determined through its surface modification properties and not by modulation sof the biological activity (Figure 14). However, several scientific questions remain to be investigated.

Firstly, the anti-adhesion results of MO245 against *P. aeruginosa* were explained by a change in the hydrophobicity of biotic and abiotic surfaces. However, concerning *V. harveyi*, MO245 led mainly to a modification of the abiotic surface. It would be interesting to demonstrate, using complementary analytical methods, the adsorption of MO245 on the surface of *V. harveyi*. Does the activity of MO245 result from a combined effect between biotic and abiotic adsorption or is there a balance between biotic and abiotic surface adsorption in order to have an anti-adhesive effect?

Secondly, concerning HA, a lower activity than MO245 was observed while it decreased the hydrophobicity of the abiotic surface in the same way. One hypothesis for this change in activity is that HA does not adsorb to surfaces in the same way as MO245 because the purity is not the same as MO245. This difference in purity could explain differences in adsorption to surfaces. The weak protein portion could also allow the adsorption of MO245 to the surfaces, allowing the polysaccharide portion to have anti-biofilm activity [38].

### 3.5. Biotechnological Interests

The EPS MO245, due to its ability to prevent adhesion, biofilm maturation, and degrade a preformed biofilm, is a new candidate as an alternative to antibiotics and biocidal agents. Moreover, the first results of its anti-biofilm mode of action could be directly linked to its surfactant properties. Due to these properties, MO245 could have a wide field of application in aquaculture, the cosmetic industry, or in the hospital environment [36].

In addition, HA, in other forms, is also often used in medical and marine applications; MO245, based on the results of this study, would make a good alternative or complement to HA.

## 4. Materials and Methods

### 4.1. Materials

The HA 1.5 MDa (protein ≤ 0.1%) was purchased from Glentham (Corsham, United Kingdom). Sodium dibasic phosphate, potassium phosphate, ammonium chloride, magnesium chloride, calcium chloride, magnesium sulfate, L-arginine, antibiotics, kojic acid, and olive oil were ordered from Sigma (Lezennes, France). Sodium chloride and glucose were bought from Grosseron (Couëron, France). Yeast extract, tryptone, agar, and glycerol were ordered from Fisher Bioreagent (Illkirch-Graffenstaden, France). Vitamin-free casamino acid was purchased from Difco (Saint-Ferréol, France). Glutaraldehyde solution, 25%, was purchased from Fisher Scientific (Illkirch-Graffenstaden, France). Triton X-100, Tris, and sodium dodecyl sulfate (SDS) were ordered from Roth (Lagny-sur-Marne, France).

### 4.2. Strains and Growth Conditions

*P. aeruginosa* PaO1 was cultivated in LB medium (NaCl 10 g/L, Yeast Extract 5 g/L, Tryptone 10 g/L) at 37 °C under agitation at 125 rpm.

*V. harveyi* DSM19623 was cultivated in Zobell medium (Sea Salt 30 g/L, Yeast Extract 1 g/L, Tryptone 4 g/L) at 28 °C under agitation at 125 rpm.

### 4.3. Industrial Production, Extraction, and Purification of MO245

Large-scale production was performed in a 500 L bioreactor containing a Zobell-adapted medium. Dextrose was added in a continuous fed batch to maintain a high C/N ratio. Aeration and agitation were maintained at the highest level, pH was adjusted at 7.4 with NaOH (8 N), and the temperature was maintained at around 30 °C. Foaming was avoided by the addition of Pluronic acid (BASF). After around 48 h cultivation, MO245 EPS was extracted from the culture medium by high-speed disk-stack centrifugation at 14,000× *g*. The supernatant was then subjected to ultrafiltration using hollow fiber, 100 kD cut-off, polyestersulfone membranes (Koch Industrie Inc., Wichita, KS, USA) and washed with deionized water. After a sterilizing frontal filtration in a 0.2 µm filter, purified EPS was concentrated up to 1% (*w*/*v*) and freeze-dried.

The purity of MO245 was moisture < 10% and protein < 3%.

### 4.4. Anti-Bacterial Activity

The anti-bacterial activities of MO245 and HA were assessed against *Pseudomonas aeruginosa* PaO1 and *Vibrio harveyi* DSM19623.

*P. aeruginosa* and *V. harveyi* were grown overnight and were then diluted to reach 10^6^ CFU/mL (O.D._600_ = 0.001) in 50 mL of culture media (LB for *P. aeruginosa* and Zobell for *V. harveyi*), containing MO245 or HA at 125, 250, or 500 µg/mL. A negative control was performed without MO245 or HA. All trials were subjected to incubation at 37 °C for 30 h under 125 rpm for *P. aeruginosa* and 28 °C for 30 h for *V. harveyi*. Every hour, the optical density (O.D._600_) was recorded at 600 nm and 0.1 mL of the dilution, between 10^1^ and 10^3^, was spread on the LB agar for *P. aeruginosa* and Zobell agar for *V. harveyi*. Agar plates were incubated at 37 °C for 24 h for *P. aeruginosa* and 28 °C for 24 h for *V. harveyi*. Unit forming colonies (UFC) were then counted.

The ability of bacteria to use MO245 or HA as a carbon source was determined. For this purpose, M9 medium (Na_2_HPO_4_ 30 g/L, KH_2_HPO_4_ 15 g/L, NH_4_Cl 0.1 g/L, and NaCl 0.5 g/L for *P. aeruginosa* and Sea Salt 20 g/L for *V. harveyi*, MgCl_2_ 2 mM, and CaCl_2_ 0.1 mM) was used. MO245 and HA were added at 125 µg/mL. A medium supplemented with glucose at 4 g/L was used as a positive control and the M9 medium alone as a negative control.

Bacteria were inoculated at 10^8^ CFU/mL (O.D._600_ = 0.1), in 50 mL M9 in 250 mL Erlenmeyer flasks, under agitation at 125 rpm, at 37 °C and 28 °C for *P. aeruginosa* and *V. harveyi*, respectively. At 3 h, 6 h, 24 h, and 26 h, the O.D. was recorded at 600 nm and dilutions between 10^1^ and 10^3^ were inoculated on LB and Zobell agar for *P. aeruginosa* and *V. harveyi*, respectively. The agar plates were then incubated at optimal temperatures. UFC were then counted.

All experiments were run in triplicate.

### 4.5. Anti-Adhesion Activity

MO245 and HA activities on the adhesion of *P. aeruginosa* and *V. harveyi* were evaluated using the flow cell system [59]. Bacterial adhesion was assessed in a three-channel flow cell (1 × 4 × 44 mm, Biocentrum DTU Danemark) prepared with a microscope glass coverslip (24 × 50 mm) [28]. Before the experiment, flow cells were sterilized under UV lamps for 30 min. The tubing system was sterilized overnight using a bleach flow at 130 µL/min and then rinsed with media (minimum medium for *P. aeruginosa* and ASW for *V. harveyi*).

Anti-adhesion activity was performed in two conditions: the first condition consisted of the addition of MO245 or HA within the bacterial suspension on the channel (Figure 4A); the second condition consisted of conditioning the surface glass slide with MO245 or HA (Figure 4B) for 2 h at room temperature, then the surface was washed 30 min at 130 µL/min. Bacteria were inoculated. MO245 and HA were used at 125 µg/mL.

For each sample, 300 µL of 10^8^ CFU/mL bacteria dilution was inoculated in every channel with a 1 mL syringe [28]. After incubation at 20 °C for 2 h, the system was washed for 30 min at 130 µL/min to remove planktonic cells.

Bacterial adhesion was observed on Confocal Laser Scanning Microscope (CLSM, LSM 710 from Zeiss, Germany) with 5 µM of Syto9^®^ nucleic acid strain (Invitrogen from fisher) by using a 40× oil immersion objective. The recovery percentage was calculated by using the ImageJ software and measuring the difference between the black and white pixels. The percentage of adhesion inhibition was represented as:(1)$$\mathrm{Percentage}\;\mathrm{adhesion}\;\mathrm{inhibition}=\left(1-\frac{\mathrm{recovery}\;\mathrm{percentage}\;\mathrm{with}\;\mathrm{MO}245\;\mathrm{or}\;\mathrm{HA}}{\mathrm{recovery}\;\mathrm{percentage}\;\mathrm{control}}\right)\times 100$$

Ten observations per condition were realized in triplicate. Those 30 acquisitions were then analyzed.

### 4.6. Impact on the Biofilm Maturation

MO245 and HA activities on the biofilm maturation of *P. aeruginosas* and *V. harveyi* were evaluated in the flow cell system. The flow cell system was prepared as described above.

*P. aeruginosa* and *V. harveyi* were adhered in the channels at 10^8^ CFU/mL for 2 h at room temperature in minimum media. The anti-biofilm activity was evaluated with a nutrient flow supplemented with MO245 or HA at 125 µg/mL at 64 µL/min for 24 h at 37 °C for *P. aeruginosa* and 28 °C for *V. harveyi*. A control without the addition of MO245 or HA in the medium was performed under the same conditions.

Bacterial biofilm was observed by CLSM as described above for bacterial adhesion. Total biomass and average thickness were determined with the COMSTAT program from MATLAB software. The normalized data were expressed as:(2)$$\mathrm{Normalized}\;\mathrm{data}=\frac{\mathrm{data}\;\mathrm{with}\;\mathrm{MO}245\;\mathrm{or}\;\mathrm{HA}}{\mathrm{data}\;\mathrm{control}}$$

The inhibition factor was calculated by a ratio between the values of biofilm between the control (bacteria without MO245 or HA) and the values obtained by the addition of MO245 or HA. The percentage of biofilm maturation inhibition was represented as:(3)$$\mathrm{Percentage}\;\mathrm{biomass}\;\mathrm{inhibition}=\left(1-\frac{\mathrm{biomass}\;\mathrm{with}\;\mathrm{MO}245\;\mathrm{or}\;\mathrm{HA}}{\mathrm{biomass}\;\mathrm{percentage}\;\mathrm{control}}\right)\times 100$$
(4)$$\mathrm{Percentage}\;\mathrm{thickness}\;\mathrm{inhibition}=\left(1-\frac{\mathrm{thickness}\;\mathrm{with}\;\mathrm{MO}245\;\mathrm{or}\;\mathrm{HA}}{\mathrm{thickness}\;\mathrm{control}}\right)\times 100$$

Seven observations per condition were realized in triplicate. Those 21 acquisitions were then analyzed.

### 4.7. Impact on the Degradation of the Biofilm

The ability of MO245 and HA to degrade a preformed biofilm was tested in a flow cell system.

*P. aeruginosa* and *V. harveyi* were adhered in the channels at 10^8^ CFU/mL for 2 h at room temperature, in minimum media. The biofilm was grown in a culture medium at 64 µL/min for 24 h at 37 °C for *P. aeruginosa* and 28 °C for *V. harveyi*. MO245 or HA at 125 µg/mL was injected in the channels with a 1 mL syringe for 2 h at 37 °C for *P. aeruginosa* and 28 °C for *V. harveyi*. A flow at 130 µL/min was added after incubation for 30 min to eliminate the degraded biofilm and planktonic cells.

Bacterial biofilm was observed by CLSM followed by image analysis as described above. The normalized data was calculated as mentioned before. The detachment factor was calculated by a ratio between the bacterial biovolume of the control (bacteria without MO245 or HA) and the bacterial biovolume obtained by the addition of MO245 or HA. The percentage of biofilm detachment was represented as mentioned above.

Seven observations per condition were realized in triplicate. Those 21 acquisitions were then analyzed.

### 4.8. Morphological Impact on Bacteria

The impact of MO245 or HA on cell morphology was determined by SEM observation. Both adhesion conditions were observed.

For the adhesion without conditioning, MO245 and HA at 125 µg/mL were incubated in the presence of bacteria at 10^8^ CFU/mL for 2 h at 20 °C. Then, the suspension was placed on a glass slide for 2 h at 20 °C. The non-adhered bacteria were washed in physiological water for *P. aeruginosa* or ASW for *V. harveyi*.

For adhesion with beforehand surface conditioning, MO245 or HA at 125 µg/mL was incubated on a glass slide for 2 h at 20 °C. Excess MO245 or HA was removed by pipetting. Bacteria at 10^8^ CFU/mL were inoculated on the conditioned glass slides for 2 h at 20 °C. The non-adhered bacteria were washed in physiological water for *P. aeruginosa* or ASW for *V. harveyi*.

All slides were fixed in a 2.5% (*v*/*v*) glutaraldehyde bath overnight at 4 °C. Afterward, the samples were washed three times for 10 min, in 0.1 M phosphate buffer with a pH of 7.35. They were then dehydrated by replacing the water with ethanol in a succession of washes: three times for 10 min in 50% ethanol, three times for 10 min in 70% ethanol, three times for 10 min in 95% ethanol, and three times for 10 min in 100% ethanol. Finally, the samples were dried in the air and then observed by SEM.

SEM was performed using a JEOL JSM-IT500HR. The samples were placed on a carbon sticker and coated with gold using a sputter coater (Scancoat6) from Edward. Observations were conducted under high vacuum conditions with an acceleration voltage of 3 kV.

### 4.9. Evaluation of the Biological Role of MO245 and HA

#### 4.9.1. Impact on the Bacterial Motility

The impact of MO245 and HA on the motility of *P. aeruginosa* and *V. harveyi* was evaluated. LB medium or Zobell medium was supplemented with 3 g/L, 5 g/L, or 10 g/L to test the impact of MO245 or HA on swimming, swarming, and twitching, respectively [29]. Bacteria were incubated at 10^8^ UFC/mL, from an overnight culture, with physiological water, MO245, or HA at 125 µg/mL for 2 h. At the end of the 2 h, plates were inoculated with sterile toothpicks. Plates were incubated overnight at 37 °C or 28 °C for *P. aeruginosa* and *V. harveyi*, respectively. Motility was analyzed by comparing the diameter of the circular zone of each condition.

All experiments were run in triplicate.

#### 4.9.2. Anti-Quorum Sensing Properties of MO245 and HA

The anti-quorum activity of MO245 and HA on *P. aeruginosa* was performed using the *E. coli* biosensor pSB401. This biosensor contained the plasmid containing the *lux*CDABE reporter gene. The plasmid pSB401 was maintained in the strain by adding tetracycline at 10 µg/mL. The addition of C6-HSL allowed the activation of the gene and the production of luminescence [60].

The strain was grown at 28 °C under 125 rpm agitation in an LB medium supplemented with 10 µg/mL tetracycline. This medium was also used for the assays. An overnight culture of *E. coli* pSB401 was inoculated into a 100 mL Erlenmeyer flask containing 20 mL of the medium at 10^8^ CFU/mL. The medium was supplemented with C6-HSL and MO245 and HA were tested at 125 µg/mL or without them as a negative control. A known quorum-sensing inhibitor, kojic acid, was added as a positive control at the same concentration as MO245 and HA [61,62].

Strains were incubated at 28 °C and a measurement of luminescence and O.D._600_ was performed every hour for 9 h using a white 96-well plate and a transparent 96-well plate, respectively (Thermo Scientific). The measurements were performed with a microplate reader (TECAN Infinite M200 pro).

The RLU was calculated to determine the impact of MO245 or HA on quorum sensing.

The impact of MO245 or HA on *V. harveyi* quorum sensing was determined via measurement of luminescence intensity over time. For this purpose, an overnight culture of *V. harveyi* was inoculated at 10^8^ CFU/mL into a 100 mL Erlenmeyer flask containing 20 mL of autoinducer (AB) medium. AB medium is composed of NaCl 0.3 M, MgSO_4_ 0.05 M, and vitamin-free casamino acids 0.2% at pH 7.5. After autoclaving, the medium was complemented with potassium phosphate 0.01 M, L-arginine 0.001 M, and glycerol 1% [63]. As described previously, MO245 and HA were tested at 125 µg/mL or without them as a negative control. A known quorum-sensing inhibitor, kojic acid, was added as a positive control [61,62].

Strains were incubated at 28 °C and a measurement of luminescence and O.D_600_ was recorded every hour for 9 h using a white 96-well plate and a transparent 96-well plate, respectively. The measurements were performed with a microplate reader (TECAN Infinite M200 pro).

The luminescence to O.D ratio was calculated to determine the impact of MO245 or HA on quorum sensing.

All experiments were run in triplicate.

### 4.10. Cell-Surface Interactions

#### 4.10.1. Microbial Adherence to Hydrocarbure (MATH)

The ability of MO245 or HA to modify the surface of bacteria was studied using MATH [30].

Pre-cultures of *P. aeruginosa* and *V. harveyi* were centrifuged for 5 min at 1000 g at room temperature. The pellet was washed in 10 mL of minimum medium (physiological water or ASW for *P. aeruginosa* and *V. harveyi*, respectively). In a 2 mL Eppendorf tube, 1 mL of bacterial culture at 10^9^ CFU/mL was placed in contact with 125 µg/mL of MO245 or HA. The solution was inoculated at room temperature for 2 h. The first measurement of O.D._600m_ was measured, namely O.D_A_. A total of 1 mL of toluene was added to the solution, then the whole was vortexed for 30 s.

The solution was then statically incubated for 1 h (in order to reach the separation phase). The O.D._600nm_ of the aqueous phase was then measured, namely O.D._B_.

The hydrophobicity of the cells was measured by the calculation:(5)$$\mathrm{Hydrophobicity}\;\%=\left(1-\frac{O.D._{B}}{O.D._{A}}\right)\times 100$$

All experiments were run in triplicate.

#### 4.10.2. QCM Measurements

MO245 and HA concentrated solutions were prepared, similarly, at a concentration of 15 g/L in Tris buffer solutions and 150 mM in NaCl stirred overnight. The pH was adjusted to 7.5. From this stock solution, diluted solutions of MO245 and HA were prepared at 750 mg/L in TRIS buffer solution at the same ionic strength and a pH of 7 by the addition of HCl (1 M). The solutions were then filtered through a 0.45 µm hydrophilic PVDF filter to eliminate any dust or residual large aggregates prior to any adsorption experiments [33].

QCM measurements were performed using a Q-sense apparatus from Biolin Scientific (Biolin, Sweden). The technique provides real-time measurements of both the resonance frequency (Δf) and energy dissipation (ΔD) of objects or materials that interact with the surface of an oscillating crystal. In this work, silica (QSX 303) coated quartz crystals, with a resonance frequency of 5 MHz, was used (Biolin, Sweden). Prior to any experiments, the crystal substrates were carefully cleaned using the following procedure: first, the substrate was submitted to UV irradiation for 10 min, then it was introduced into a sodium dodecyl sulfate (SDS) solution (2%) for 30 min, rinsed thoroughly MilliQ water, and dried under filtered N2 flux. Finally, the dry substrate was again placed under UV light for 10 min and directly introduced in the QCM flow-through cell. Adsorption experiments were performed by first stabilizing the frequency and dissipation signal in a blank solution (buffer solution without MO245 or HA) for at least 30 min under ambient temperature. Once a stable baseline was obtained, the solution containing the MO245 or HA was injected at a volume rate of 100 µL/min at ambient temperature. At the end of the adsorption time, the system was rinsed again with the buffer solution free of MO245 or HA. The QCM measurements were recorded at several overtones (*n* = 3, 5, 7, 9 11). For each sample, a minimum of three replicas were performed to ensure a good reproducibility in the adsorption measurements and cleaning procedure of both substrates and QCM cells.

#### 4.10.3. Cell-Surface Interactions

Surface modification by EPS was calculated by contact angle measurements. Glass coverslips were conditioned with MO245 or HA at 125 µg/mL. Briefly, 3 mL of MO245 and HA were deposited on a glass slide and allowed to evaporate in a sterile environment. The contact angles were then measured with ultrapure water. Water contact angles were measured with Digidrop (Digidrop, GBX, UK) at room temperature. The volume used was 3 µL.

Five measurements on three slides were performed and all values were averaged. The resulting 15 measurements were then analyzed.

#### 4.10.4. Emulsifying Properties

The emulsifying activities of MO245 or HA were determined by the emulsion index [64]. In a 1.5 cm glass tube, 1.5 mL of 0.25% (*w*/*v*) MO245 or HA was added to 2.25 mL of olive oil. The solution was then stirred for 2 min at 40 Hz. Triton X-100 was used as a known reference surfactant [35]. After 1 h, 24 h, 48 h, and 168 h at 20 °C, the emulsion index was calculated as:(6)$$E\;\%=\frac{H_{e}}{H_{t}}\times 100$$

With H_e_ as the height of the emulsion layer and H_t_ as the total height of the mixture.

All the tests were performed in triplicate.

### 4.11. Statistical Analysis

All data were statistically analyzed with R studio software. Two-by-two comparative of means were performed using the Mann–Whitney test. The significant level alpha was set at 0.05.

## Figures and Tables

**Figure 1 marinedrugs-20-00728-f001:**
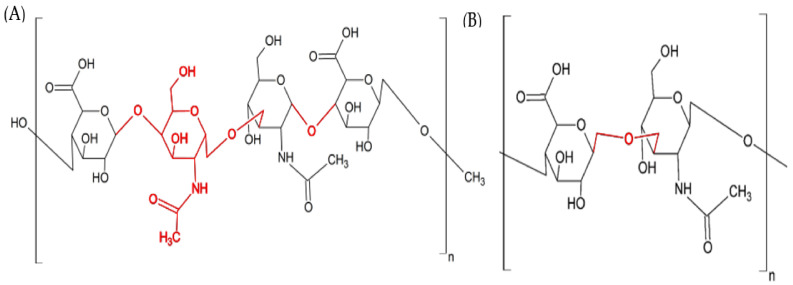
Structure of (**A**) MO245 and (**B**) HA.

**Figure 2 marinedrugs-20-00728-f002:**
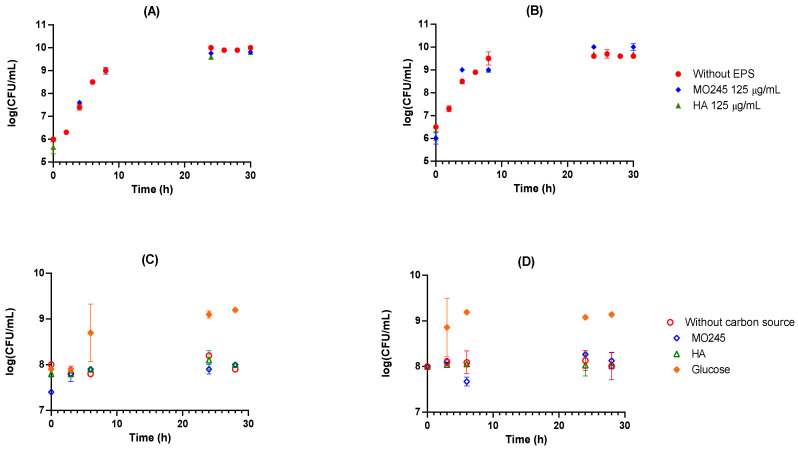
Impact of MO245 or HA (125 µg/mL) on the growth of (**A**) *P. aeruginosa* in LB medium at 37 °C and (**B**) *V. harveyi* in Zobell medium at 28 °C for 30 h under 125 rpm agitation. Average of three independent replicates ± standard deviation. Bactericidal effect and consumption of MO245 or HA (125 µg/mL) or glucose (4 g/L) as a carbon source in M9 medium for (**C**) *P. aeruginosa* at 37 °C and (**D**) *V. harveyi* at 28 °C for 26 h under 125 rpm agitation.

**Figure 3 marinedrugs-20-00728-f003:**
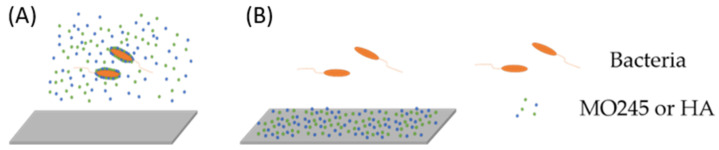
Conditions of use of MO and HA during adhesion. (**A**) Addition of MO245 or HA within the bacterial suspension and (**B**) glass slide conditioned with MO245 or HA.

**Figure 4 marinedrugs-20-00728-f004:**
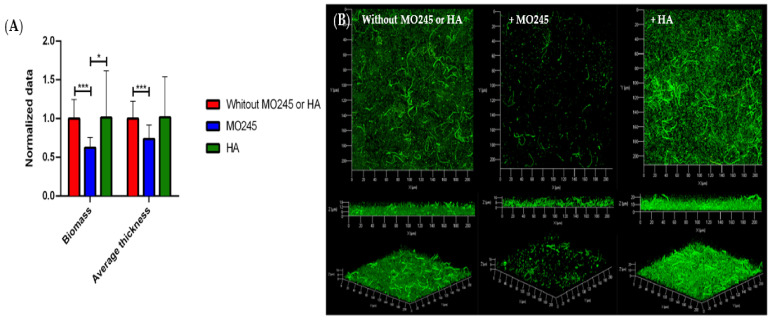
Impact of MO245 and HA on a 24 h *P. aeruginosa* biofilm maturation. (**A**) Biomass and average thickness quantification after COMSTAT analysis of confocal laser microscopy observations. (**B**) Confocal laser microscopy observation (Syto9^®^) without or with the addition of MO245 or HA at 125 µg/mL in the LB growth medium. Data represent the mean ± the standard deviation. * represents the significant difference at α 5%: *p* < 0.05, *** represents the significant difference at α 5%: *p* < 0.001.

**Figure 5 marinedrugs-20-00728-f005:**
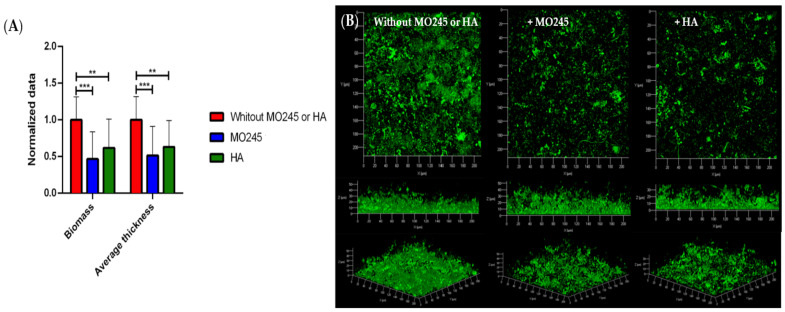
Impact of MO245 and HA on a 24 h *V. harveyi* biofilm maturation. (**A**) Biomass and average thickness quantification after COMSTAT analysis of confocal laser microscopy observation. (**B**) Confocal laser microscopy observation (Syto9^®^) without or with the addition of MO245 or HA at 125 µg/mL in the Zobell growth medium. Data represent the mean ± the standard deviation. ** represents the significant difference at α 5%: *p* < 0.01; *** represents the significant difference at α 5%: *p* < 0.001.

**Figure 6 marinedrugs-20-00728-f006:**
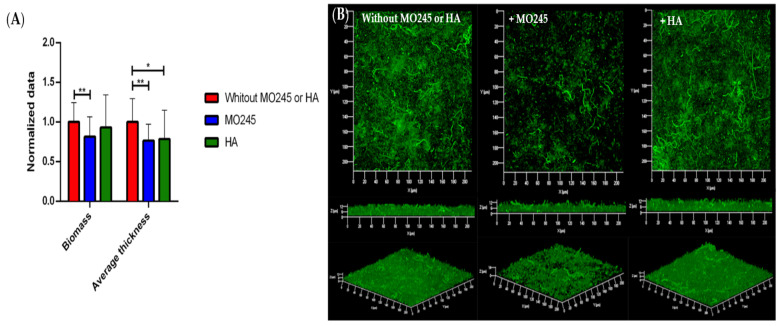
Impact of MO245 and HA on the degradation of a 24 h *P. aeruginosa* biofilm already formed. (**A**) Biomass and average thickness quantification after COMSTAT analysis of confocal laser microscopy observation. (**B**) Confocal laser microscopy observation (Syto9^®^) without or with the addition of MO245 or HA at 125 µg/mL on a 24 h biofilm already formed. Data represent the mean ± the standard deviation. * represents the significant difference at α 5%: *p* < 0.05, ** represents the significant difference at α 5%: *p* < 0.01.

**Figure 7 marinedrugs-20-00728-f007:**
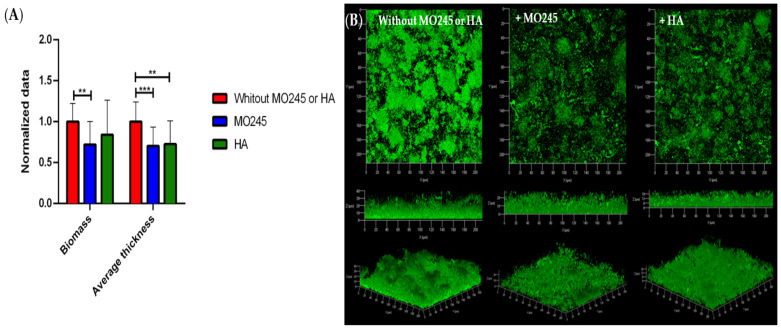
Impact of MO245 and HA on the degradation of a 24 h *V. harveyi* biofilm already formed. (**A**) Biomass and average thickness quantification after COMSTAT analysis of confocal laser microscopy observation. (**B**) Confocal laser microscopy observation (Syto9^®^) without or with the addition of MO245 or HA at 125 µg/mL on a 24 h biofilm already formed. Data represent the mean ± the standard deviation. ** represents the significant difference at α 5%: *p* < 0.01, *** represents the significant difference at α 5%: *p* < 0.001.

**Figure 8 marinedrugs-20-00728-f008:**
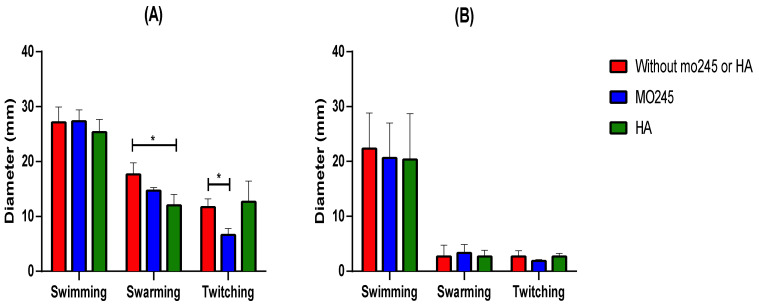
Motility assay of (**A**) *P. aeruginosa* and (**B**) *V. harveyi* incubated with MO245 or HA at 125 µg/mL or nothing for 2 h at room temperature. Diameters were measured after the overnight incubation of agar plates at 37 °C for *P. aeruginosa* and 28 °C for *V. harveyi*. Data represent the mean ± the standard deviation. * represents the significant difference at α 5%: *p* < 0.05.

**Figure 9 marinedrugs-20-00728-f009:**
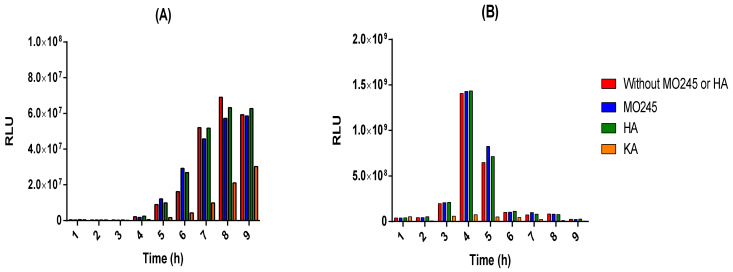
Anti-quorum sensing effect of MO245, HA, and kojic acid (KA) at 125 µg/mL on (**A**) the biosensor *E. coli* pSB401 and (**B**) *V. harveyi* for 9 h at 28 °C under 125 rpm agitation. Luminescence and O.D._600_ were measured every hour and RLU ratios were calculated.

**Figure 10 marinedrugs-20-00728-f010:**
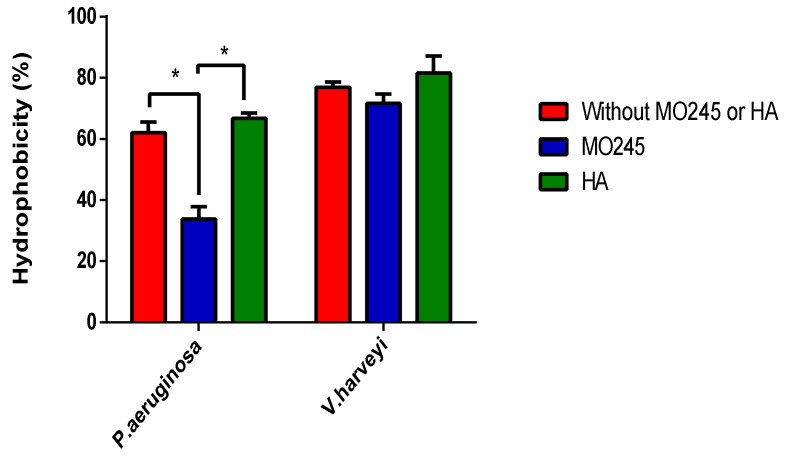
Percentage hydrophobicity of *P. aeruginosa* and *V. harveyi* in the presence of MO245 or HA. A total of 10^9^ bacteria were put in contact with MO245 or HA at 125 µg/mL for 2 h and then in contact with toluene. At the appearance of a phase separation, the aqueous phase was recovered and the optical density was measured. The percentage of hydrophobicity was then calculated. * represents the significant difference at α 5%: *p* < 0.05.

**Figure 11 marinedrugs-20-00728-f011:**
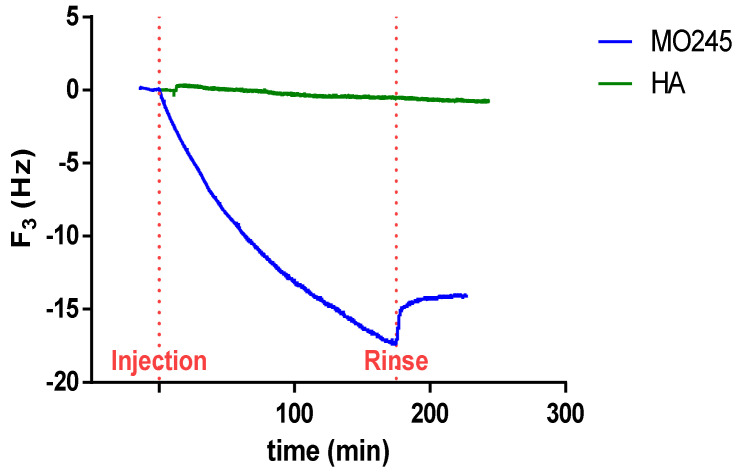
Evolution of the frequency F3 versus time for MO245 and HA on silica-coated quartz crystal (pH 7, 150 mM NaCl). MO245 and HA were injected at t = 0 min. After the adsorption of MO245 on the surface of silica-coated quartz crystal, rinsing was performed to observe if desorption occurred.

**Figure 12 marinedrugs-20-00728-f012:**
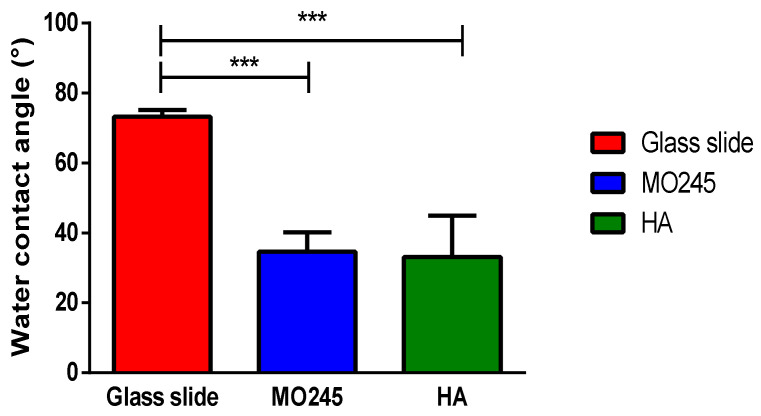
The water contact angle of a surface conditioned with MO245 or HA at 125 µg/mL measured with a Digidrop. A total of 3 mL of MO245 and HA were deposited on a glass slide and allowed to evaporate in a sterile environment. Contact angles were measured at room temperature with a volume of 3 µL. *** represents the significant difference at α 5%: *p* < 0.001.

**Figure 13 marinedrugs-20-00728-f013:**
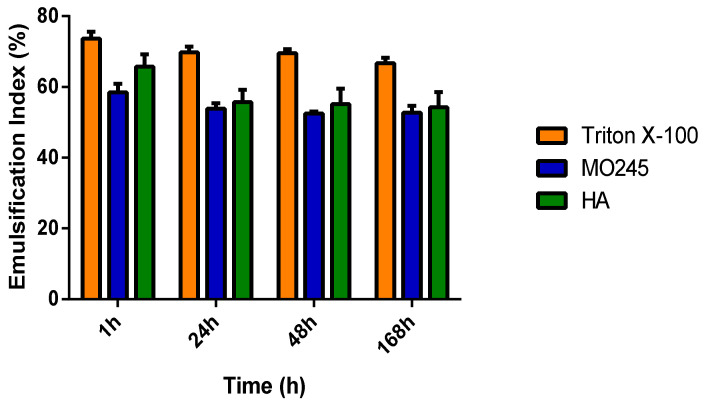
Emulsifying properties of MO245, HA, and Triton X-100 at 0.25% (*w*/*v*) in the oil-in-water phase over time. The emulsion index was calculated after 1 h, 24 h, 48 h, and 168 h at 20 °C.

**Figure 14 marinedrugs-20-00728-f014:**
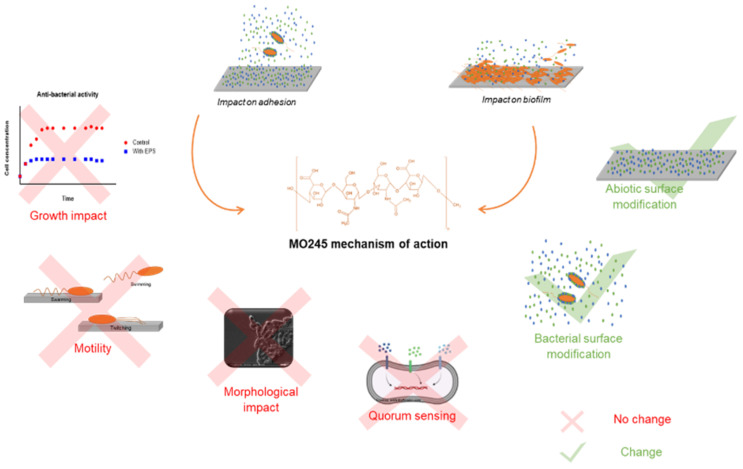
Summary scheme of the potential anti-biofilm mode of action of MO245.

**Table 1 marinedrugs-20-00728-t001:** Comparison of the known rheological properties of MO245 with the properties of HA described in the literature.

Polysaccharide	M_W_ (Da)	Viscosity (mPa·s)	G′ (Pa) ^(a)^	G″ (Pa) ^(a)^	Ref
MO245	1.5 × 10^6^	5	0.9	20	[21]
HA	1.1 × 10^6^	11.6	55.8	67.5	[27]
	2.0 × 10^6^	107	220	125	

^(a)^: Frequency = 2.5 Hz.

**Table 2 marinedrugs-20-00728-t002:** Anti-adhesion activity of MO245 and HA at 125 µg/mL on *P. aeruginosa* and *V. harveyi* under two conditions.

Strain	Percentage of Adhesion Inhibition			
	First Condition		Second Condition	
	MO245	HA	MO245	HA
*P. aeruginosa*	−53 ± 15% ***	−6 ± 11%	−44 ± 15% ***	+6 ± 12%
*V. harveyi*	−29 ± 18% ***	0 ± 15%	−49 ± 17% ***	−4 ± 24%

*** Significant difference at α 5%: *p* < 0.001 with the control without MO245 or HA.

## Data Availability

Not applicable.

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
