# Peer review of "Anti-Biofilm Activity of a Hyaluronan-like Exopolysaccharide from the Marine Vibrio MO245 against Pathogenic Bacteria"

_marinedrugs, 2022, doi:10.3390/md20110728_

Round 1
Reviewer 1 Report
The authors investigated the antibiofilm activity of a hyaluronan-like exopolysaccharide from Vibrio MO245 via multiple aspects including anti-bacterial, anti-adhesion, degradation of the formed biofilm as well as the physicochemical property of MO245. Overall, the results were sufficiently presented and discussed in the study. Nevertheless, several doubtful points should be clarified.
Q1. Line 121, from the results, it was suggested that three concentrations (125, 250, and 500) were tested during the experiment and the dose of 125 μg/mL was selected for the following investigation. Please specify the reason in the manuscript.
Q2. The superscript 1 in Table 2 should better be unified with other tables and expressed as “*”. Besides, the representation of “*” should better be unified (e.g. Line 189: no significant impact p<0.05).
Q3. The full names of abbreviations (e.g. QCM, ASW) should be provided in their first appearance. Other abbreviations should better be checked carefully, please.
Q4. Line 488, the information of microorganisms for the production of MO245 should be better provided. Besides, was the 100 kDa cut-off ultrafiltration procedure possible to prepare MO245? More details should better be provided, please.
Q5. The potential mechanism for the antibiofilm activity of MO245 should be summarized in a scheme figure in the discussion part.
Q6. Form the results, it seemed that the physicochemical property of MO245 was more dedicated to its antibiofilm activity. Was it possible to design a specific antibiofilm agent based on such a mechanism? Since HA showed little differences from MO245, was it possible to modify HA to achieve similar effects?
Reviewer 2 Report
The manuscript is well written and the experiments are coherent. Results are properly described and the results are clear.
A minor point is in line 496: against will be more adequate than toward
However, if the authors are stating that MO245 could be useful in clinics, they should address at least in the discussion, some of the following questions:
.To show the utility in clinical devices, the authors have to show its absence of toxicity on animal cells. And that MO245 does not liberates itself from the device and reaches the internal fluids of the body.
. With respect to the effect on the inhibition of the adhesion, how long does it last? Because it has to last for long periods of time either in an aquatic environment or in body fluids, for instance.
. Have the authors assayed the inhibition capacity on catheteres, or dental devices? It would be very helpful for the clinical use of the polymers.
.The authors have used bacterial models, have they tried the same experiments with Candida albicans? Do they have references with this model with other EPSs?
Reviewer 3 Report
This manuscript describes a polysaccharide extracted from marine bacterium on the growth, adhesion, biofilm maturation and dispersion of two pathogenic model strains, Pseudomonas aeruginosa sp. PaO1 and Vibrio harveyi DSM 19623.
1- The manuscript contains many figures that could be compressed for better fluidity. Figures 2 and 3 can be combined into one (A-D), Figure 4, for example, could be in Supplementary material or Material and Methods. The psycho-chemical impact of the polysaccharide (Figs 11-14) can also be compressed into one figure.
2- Figure legends, in general, should contain more information on how the experiment was performed.
3- Authors should better explain the “quorum sensing” experiment. For those who are not specialists, it is difficult to understand what is being measured in the experiment.
4- Authors must correct spelling and grammar errors in the manuscript.
Round 2
Reviewer 1 Report
The authors have clarified the concerns about the previous manuscript.